# The Biosynthesis and Functions of Polyamines in the Interaction of Plant Growth-Promoting Rhizobacteria with Plants

**DOI:** 10.3390/plants12142671

**Published:** 2023-07-17

**Authors:** Michael F. Dunn, Víctor A. Becerra-Rivera

**Affiliations:** Programa de Genómica Funcional de Procariotes, Centro de Ciencias Genómicas, Universidad Nacional Autónoma de México, Cuernavaca 62210, Mexico; cate88_6@hotmail.com

**Keywords:** rhizosphere signaling, plant growth-promoting rhizobacteria, rhizomicrobiome, polyamines, abiotic stress resistance

## Abstract

Plant growth-promoting rhizobacteria (PGPR) are members of the plant rhizomicrobiome that enhance plant growth and stress resistance by increasing nutrient availability to the plant, producing phytohormones or other secondary metabolites, stimulating plant defense responses against abiotic stresses and pathogens, or fixing nitrogen. The use of PGPR to increase crop yield with minimal environmental impact is a sustainable and readily applicable replacement for a portion of chemical fertilizer and pesticides required for the growth of high-yielding varieties. Increased plant health and productivity have long been gained by applying PGPR as commercial inoculants to crops, although with uneven results. The establishment of plant–PGPR relationships requires the exchange of chemical signals and nutrients between the partners, and polyamines (PAs) are an important class of compounds that act as physiological effectors and signal molecules in plant–microbe interactions. In this review, we focus on the role of PAs in interactions between PGPR and plants. We describe the basic ecology of PGPR and the production and function of PAs in them and the plants with which they interact. We examine the metabolism and the roles of PAs in PGPR and plants individually and during their interaction with one another. Lastly, we describe some directions for future research.

## 1. Introduction

Human population increase, environmental degradation, and climate change pose significant challenges to sustainable food crop production. It is essential that the yield of crops is increased while minimizing negative impacts on the environment and ensuring economic sustainability. Selective plant breeding has increased crop productivity with the trade-off that the resulting varieties require substantial inputs of mineral fertilizers and pesticides to deliver high yields. Both traditional breeding and genetic engineering approaches for crop plant improvement have the disadvantages of long development times, technical limitations, and potential problems with public acceptance (reviewed in [1,2,3]).

An alternative approach to increasing plant productivity relies not on their genetic modification but on promoting their growth using plant growth-promoting microbes (PGPM) [4]. PGPM are part of the soil microbiome, which includes both microbes in the soil and their metabolic or structural elements, such as polysaccharides and secondary metabolites. PGPM species are adapted to interact with plant species to form relatively species-specific plant rhizosphere communities or rhizomicrobiomes. The PGPM component of the rhizomicrobiome includes mycorrhizal fungi, plant growth-promoting archaea, and plant growth-promoting rhizobacteria (PGPR). The PGPR are the most numerous members of the PGPM and enhance plant growth and stress resistance by increasing nutrient availability to the plant, producing phytohormones, stimulating plant defense responses against abiotic stresses and pathogens, and/or fixing nitrogen. These and other mechanisms underlying plant growth promotion by PGPR have been extensively reviewed [1,5,6,7,8,9,10,11] and will not be treated in detail here.

Much of the ability of PGPR to increase plant growth derives from their ability to modify plant root architecture and growth and thereby increase nutrient uptake [12]. These changes in the roots can be mediated by phytohormones (auxins, cytokinins, gibberellins) produced by PGPR, or by PGPR manipulating plant ethylene levels. PGPR affect gene expression for signaling and developmental pathways in plants, but the chemical signals involved are largely unknown [9]. Some PGPR protect plants against pathogens by niche competition, producing antibiotics and siderophores, and stimulating systemic defense systems against pathogens. Diazotrophic PGPR use the enzyme nitrogenase to reduce (fix) biologically inert atmospheric nitrogen to biologically usable ammonia. Reduced nitrogen is often the limiting nutrient in agriculture, and so diazotrophs are of great practical importance in the nitrogen nutrition of crop plants as well as having other plant-growth-promoting activities. For general reviews on PGPR in agriculture, see [1,11,13,14,15,16,17,18].

The presence of genes encoding plant growth-promoting traits in a given bacterium does not mean that it will promote plant growth. This stems from the differing degrees of plant–PGPR specificity and the existence of unknown factors involved in chemical signaling and nutrient exchange between the partners that result in growth promotion [9,19,20]. The complexity of interactions between different PGPR in synthetic communities (symcoms) that were inoculated onto corn was evident from the very dissimilar abilities of different syncoms to increase plant growth and disease resistance [21].

The gains in plant health and productivity brought about by PGPR have long shown practical benefits in agriculture through the application of commercial PGPR inoculants to certain crops (reviewed in [1,6,8,11,22]). The ability of PGPR inoculants to increase crop yields has been uneven in practice and the potential of the technology is underrealized. To overcome present limitations in PGPR efficiency and application, we need to learn more about how different PGPR species interact among themselves and with a variety of crop species under diverse environmental conditions. This knowledge can then be applied to developing more effective and reliable PGPR inocula, including formulations containing syncoms assembled using systems biology approaches [4,21,23].

For the establishment of plant–PGPR relationships, the bacteria best adapted to utilize the nutrients present in root exudates are generally the most successful at colonizing the roots. The root exudate composition thus shapes the microbial composition of the rhizosphere [20,24]. Well known examples of compounds involved in chemical communication include legume flavonoids, rhizobial nodulation factors, exopolysaccharides (EPS), and phytohormones produced by both plants and PGPR [8,15,17,25,26,27].

Polyamines (PAs) are primordial polycations with important or essential functions in all living things, such as physiological effectors and signal molecules in plant–microbe interactions. Pas are short hydrocarbons with two or more amino groups that are positively charged at physiological pH. The positive charge on PAs lets them associate with negatively charged macromolecules, such as nucleic acids, phospholipids, and some proteins, in the cell. These associations mediate cell physiology by changing the chemical or physical properties of the macromolecules with which they interact or by directly regulating their biological activities. PA catabolism generates reactive nitrogen and oxygen species that serve as signal molecules, and PAs can also be used by bacteria as sources of carbon and/or nitrogen [28]. 

The PAs commonly found in organisms include the diamine putrescine (H_2_N-(CH_2_)_4_-NH_2_), the triamine spermidine (H_2_N-(CH_2_)_3_-NH-(CH_2_)_4_-NH_2_), and the tetraamine spermine (H_2_N-(CH_2_)_3_-NH-(CH_2_)_4_-NH-(CH_2_)_3_-NH_2_). Spermidine (Spd) has an essential role in eukaryotes and archaea because it is required for the hypusine chemical modification of their IF5A translation factors. In eukaryotes, PAs are also important in cell growth and proliferation, while in archaea, less is known about their functions apart from their role in translation. PAs in bacteria can either be produced endogenously or transported from the environment. General reviews on PA metabolism and functions include [29,30,31,32].

Bacteria are especially versatile in their ability to synthesize many kinds of PAs. This may be because PAs in bacteria are involved in diverse processes, including growth, motility, gene expression, and biofilm formation [29,30,33,34,35]. PAs also condition pathogenic and mutualistic interactions between a wide range of bacteria and their animal or plant hosts (reviewed in [25,30,32,34,35,36,37,38,39,40,41]. In this review, we describe the production and function of PAs in PGPR and the plants with which they interact. In Section 2, we look at the basic ecology of PGPR, with an emphasis on those species that fix nitrogen in association with plants. In Section 3 and Section 4, we examine the metabolism and the roles of PAs in PGPR and plants, respectively. How PAs act as signal molecules and physiological effectors in the interaction of PGPR and plants and conclusions and perspectives are the subjects of Section 5 and Section 6. 

## 2. The Rhizomicrobiome

The rhizosphere is broadly defined as the soil surrounding and chemically or physically affected by the root, including root tissues that are colonized by microbes [8]. The terms endorhizosphere, rhizoplane, and ectorhizosphere describe the spatial proximity and the intimacy of the interaction of microbes with the roots (Figure 1). The endorhizosphere is the innermost rhizosphere zone and is made up of the apoplastic (extracellular) fluid-filled spaces around root cells as well as their interior. The rhizoplane refers to the surface of the root, including mucilage. The ectorhizosphere begins at the outer limit of the rhizoplane and extends a short distance (millimeters) into the bulk soil [42] (Figure 1). Distinct species of free-living, associative, and endosymbiotic PGPR inhabit one or more of these rhizospheric niches [43].

Many species of free-living bacteria live in the bulk soil, but their greatest populations occur in the rhizoplane and ectorhizosphere, where root exudates act as selective chemoattractants and nutrients [14,24,44]. Free-living bacterial diazotrophs occur in the phyla Firmicutes, Cyanobacteria, and Proteobacteria. Rhizosphere-dwelling *Azospirillum*, *Azotobacter,* and cyanobacteria (mostly *Anabaena* and *Nostoc*) species increase plant growth by fixing nitrogen and/or producing auxin [17,45]. All of the diazogrophic archaea are methanogens in the phylum Euryarchaeota [20,26].

Associative diazotrophs have facultative interactions with plants and live in the rhizoplane or endophytically in the endorhizosphere. These microbes partner with a wide variety of plants, including cereals, ferns, and bryophytes (non-vascular plants) [46]. The root-colonizing alpha-proteobacterium *Rhodopseudomonas palustris*, for example, promotes the growth of Chinese cabbage by increasing auxin levels in leaves and nitrate uptake by roots [19]. Rhizoplane and endorhizospheric diazotrophic PGPR can improve plant growth by fixing nitrogen, increasing soil nutrient availability and/or uptake, secreting phytohormones and enhancing plant pathogen resistance by niche competition, or stimulating plant resistance responses [27]. Genome sequence analysis revealed genes for all of these traits in the rhizosphere bacterium *Klebsiella variicola* D5A. Genes for PA production were defined among those that confer PGPR fitness in this strain [47,48].

Endophytic nitrogen fixers like *Azoarcus* and *Nostoc* (Pseudomonadata and Cyanobacteria, respectively) directly access plant tissues via fissures at the sites where lateral roots emerge or through other natural openings. Diazotrophic endophytes remain outside of plant cells, fix nitrogen and contribute other PGPR activities to enhance the growth in crops like wheat, rice and corn [16,45]. Despite the tight association of endophytic species with plants, they do not damage plant tissues or elicit defense responses [46].

Intracellular diazotrophic endophytes reside inside root cells and fix atmospheric nitrogen for their hosts. Rhizobia is the collective term for bacteria that initiate a chemical dialogue with selected (mostly leguminous) hosts and induce the formation of specialized organs called nodules on the roots. Metabolically differentiated rhizobia living inside of plant cells of the nodule are called bacteroids, and they receive photosynthate-derived carbon sources from the host in exchange for reduced nitrogen. Rhizobia include alphaproteobacteria that elicit the formation of root nodules on legumes or the non-legume *Parasponia* and include species of *Rhizobium*, *Sinorhizobium*, *Mesorhizobium*, *Bradyrhizobium* and *Allorhizobium* [46,49,50]. Rhizobia belonging to the beta-proteobacteria include species in the genera *Burkholderia* and *Cupriavidus*, which nodulate *Mimosa* spp. and a few other legumes [51]. The Gram-positive actinobacterial genus *Frankia* forms nitrogen-fixing nodules on some actinorhizal plants and also fixes nitrogen in free life, unlike the majority of nodule-forming rhizobia. *Azorhizobium caulinodans* (phylum Pseudomonadota) forms both stem and root nodules on *Sesbania rostrata* and can also fix nitrogen asymbiotically [49]. In addition, some *Rhizobium* species are intercellular endophytes that reside in the stems of non-legumes like cottonwood trees and are thought to fix nitrogen as members of the PGPR consortia found there [52,53].

Globally, endosymbiotic rhizobia fix more nitrogen than free-living or associative diazotrophs. Many legumes are important crops for human consumption or fodder or are used in crop rotation with nonlegumes to provide a partial replacement for chemical nitrogen fertilizer [11]. Rhizobia also promote the growth of non-legumes, such as rice and radish, by mechanisms other than nitrogen fixation, including siderophore production, phosphate solubilization and indole-3-acetic acid (IAA) production [54,55]. Non-rhizobial PGPR inoculated onto legumes in combination with compatible rhizobia can increase nodulation, nitrogen fixation, and crop yields [56,57]. 

Several genera of archaeal methanogens promote plant growth by mechanisms similar to those of the bacterial PGPR [18]. Methanogens that possess the gene encoding the iron protein of nitrogenase (*nifH*) are presumed to be nitrogen fixers and are common in many marine environments and wetlands [58]. Much less is known about their interactions with plants than for bacterial PGPR [26].

## 3. Polyamine Metabolism and Functions in PGPR

In this section, we consider the metabolism (Figure 2) and physiological functions of PAs in PGPR. Later sections will describe these aspects in plants and in PGPR–plant interactions. 

### 3.1. Polyamine Biosynthesis in PGPR

The diamine putrescine (Put) is made by the decarboxylation of L-arginine (Arg) or L-ornithine (Orn) by pyridoxal 5′-phosphate dependent decarboxylases. Put is produced in one step by the decarboxylation of Orn by the enzyme Orn decarboxylase (Odc, EC 4.1.1.17) or from Arg by one of two pathways using Arg decarboxylase (Adc, EC 4.1.1.19). In the first of these pathways, Arg decarboxylation by Adc produces agmatine, which is converted to Put and urea by agmatinase (EC 3.5.3.11) (Figure 2). In the second pathway, agmatine produced by Adc is hydrolyzed to *N*-carbamoylputrescine by agmatine deiminase (EC 3.5.3.12), and this intermediate is converted to Put by *N*-carbamoylputrescine amidase (EC 3.5.1.53) [33] (Figure 2).

Whether bacteria exclusively or principally use Odc or Adc to make Put may largely depend on which basic amino acid substrate was most available during their evolution [59]. In PGPR, Put production by Odc appears to be the more common route. This is true in nodule-forming rhizobia and in the sugarcane endophyte *Gluconacetobacter diazotrophicus* PAI 5 [36,60,61,62]. In contrast, the endophytic *Klebsiella* spp. LTGPAF-6F and D5A and the heterocyst-forming cyanobacterium *Anabaena* sp. PCC 7120 use Adc and agmatinase to produce Put from Arg [47,63,64]. Genome analysis of the endophytes *Klebsiella pneumoniae* 342 (Enterobacteriaceae) and *Methylobacterium* P1–11 (Methylobacteriaceae) and the endosymbiotic *Tardiphaga* P9–11 (Bradyrhizobiaceae) show both Adc and Odc as potential enzymes for Put biosynthesis [65,66].

Orn is an intermediate in the Arg biosynthetic pathway, where it is formed by the deacetylation of *N*-acetylornithine. The *Medicago* endosymbiont *S. meliloti* produces several non-homologous *N*-acetyl-ornithine deacetylase paralogs to produce Orn. Orn is also produced by the hydrolysis of Arg by arginase ArgI1 [67,68,69]. In *S. meliloti* Rm8530, about 90% of the free intracellular Put is made by a L/Odc designated Odc2 [60], with the remainder being of unknown origin. It seems that metabolism in *S. meliloti* is geared towards the production of Orn for making Put as well as Orn-based siderophores and Orn-containing lipids [69]. 

Homologs of the *S. meliloti* Odc2 occur in other species of nitrogen-fixing rhizobia and in *A. tumefaciens* [36]. The Odc activity of the *S. meliloti* Odc2 is ten-fold higher than its Ldc activity, and *odc2* mutants of strain Rm8530 produce very low levels of Put, homospermidine (HSpd) and Spd [60]. An additional Odc in *S. meliloti*, called Odc1, is specific for Orn as a substrate but has very low specific activity compared to Odc2. The inactivation of *odc1* does not alter PA production under any growth condition tested [60]. 

The diamine cadaverine (Cad) is produced by Ldcs that are specific for L-lysine (Lys) or by L/Odcs [33,70,71]. Ldcs are present in some diazotrophic actinobacteria, firmicutes, and proteobacteria [36,72]. Among associative or free-living diazotrophs, *Azotobacter* spp., including *A. vinelandii*, have high levels of Put and lower levels of Cad and Spd in cells from minimal medium cultures. Low amounts of 1,3-diaminopropane (DAP) were detected in less than one-third of the strains analyzed [73,74]. The *Azosprillum brasilense* PGPR commercial inoculant strains Az39 and Cd produced and excreted high levels of Spd (>100 nmoles/mL) and 5–20 nmoles/mL of Put, spermine (Spm) and Cad when grown in minimal medium. When the supernatants of cultures grown with exogenous Lys were analyzed, Spd content was lowered by more than 90%, and Cad increased about 3-fold. The increase in Cad production when the medium was supplemented with the Ldc substrate, Lys, is not unexpected. How Lys decreases Spd biosynthesis (via Spd synthase), increases its catabolism or export is not known [75,76] (Figure 2). Most of the Cad produced by *S. meliloti* Rm8530 is linked to macromolecules rather than being exported or present in free form in the cytoplasm [60]. 

The triamine Spd is made by one of two different pathways (Figure 2). In the canonical pathway, Spd synthase (EC 2.5.1.16) transfers the aminopropyl group from decarboxylated S-adenosylmethionine (dcSAM) to Put to produce Spd. The dcSAM required for this reaction is produced by S-adenosylmethionine decarboxylase (SAMdc, EC 4.1.1.50).

The canonical pathway exists in some bacteria and in many eukaryotes. The *Klebsiella* spp. LTGPAF-6F and D5A and *Gluconacetobacter diazotrophicus* Pal 5 are diazotrophic endophytes whose genomes encode the canonical Spd pathway enzymes SAMdc (SpeD) and Spd synthase (SpeE) [47,61,64]. 

The second, or “alternative”, Spd pathway converts Put and *L*-aspartate ß-semialdehyde to carboxyspermidine using carboxynorspermidine dehydrogenase (CANSDH, EC 1.5.1.43). Carboxyspermidine is then decarboxylated by carboxynorspermidine decarboxylase (CANSDC, EC 4.1.1.96) to produce Spd (Figure 2). This pathway is found in many species of Bacteroidetes, Firmicutes, and proteobacteria [34]. Put and Spd are the predominant intracellular pAs in the associative species *Azoarcus rhizosphaerae*, a nitrogen-fixing member of the *Fiscus religiosa* (sacred fig) rhizomicrobiome [77]. PA screening of a collection of 15 endophytic actinorhizal isolates from mangrove roots showed that two strains with nitrogenase activity produced Put, Spd, and the tetraamine Spm [78]. 

To make Spm, Spm synthase (EC 2.5.1.22) transfers the aminopropyl group from dcSAM to Spd. Thermospermine (TSpm), the structural isomer of Spm, is made in plants from the same substrates in a reaction catalyzed by TSpm synthase (EC 2.5.1.79) [33,79] (Figure 2).

*S. meliloti* produces Spd by the “alternative” pathway described above. Neither this pathway nor the canonical Spd synthase pathway is present in most other alpha-proteobacterial rhizobia [36,60,80], consistent with their lack of Spd production [36,81]. 

Essentially all rhizobia and many other PGPR produce the triamine HSpd, which is formed by the condensation of two Put molecules using homospermidine synthase (Hss, EC 2.5.1.44) [81,82] (Figure 2). The Hss enzymes of *B. japonicum*, *R. leguminosarum* bv. viciae and *S. meliloti* can also use Spd and Put as co-substrates to produce HSpd and DAP ([36,82,83], Dunn, unpublished). Strains of *Frankia* are unusual in that they do not produce HSpd, but instead synthesize mostly Put and lower amounts of Cad, Spd, and Spm [84]. 

In contrast to *S. meliloti*, most legume-nodulating rhizobia produce only Put and HSpd when grown in cultures. HSpd is often the major PA in free-living rhizobia and bacteroids [36,81,85]. The nodule-forming betaproteobacterial genera *Burkholderia* and *Cupriavidus* usually contain 2-hydroxyputrescine (hydroxy-Put) and Put rather than Put and HSpd [81,85,86]. Hydroxy-Put biosynthesis in these diazotrophic beta-proteobacteria may result from the action of Put hydroxylases like those of the non-diazotrophic ß-proteobacteria *Bordetella bronchiseptica* and *Ralstonia solanacearum*. Hydroxy-Put may function in iron binding, as it forms part of the structure of the alcaligin siderophore produced by some ß-proteobacterial species [87]. 

HSpd is made by some free-living and associative diazotrophs, including the cyanobacterium *Anabaena* [30,63,88]. In *Anabaena* sp. PCC 7120, the enzyme SpeY converts Put to HSpd. SpeY is a homolog of eukaryotic and archaeal deoxyhypusine synthase, required for using Spd to post-translationally modify the IF5A translation factor [63]. In the associative diazotrophs belonging to *Azospirillum* spp., Put, Spd, and HSpd are major intracellular PAs in cells grown in culture [81,86]. 

In addition to producing Put, Spd and HSpd, *S. meliloti* makes the triamine norspermidine (NSpd) (Figure 2). To initiate the NSpd biosynthesis pathway glutamate and L-aspartate ß-semialdehyde are converted to DAP by the sequential activities of diaminobutyric acid (DABA) aminotransferase (DABA AT, EC 2.6.1.76) and DABA decarboxylase (DABA DC, EC 2.6.1.76) [37,60] (Figure 2). NSpd is produced from DAP and L-aspartate ß-semialdehyde by the consecutive actions of CANSDH and CANSDC, the same enzymes that participate in the alternative Spd pathway [37,60,71,89] (Figure 2). In *S. meliloti* Rm8530 NSpd is principally bound to macromolecules rather than being in free form [60]. 

Genome sequences of diazotrophic PGPR often encode multiple putative transport systems to take up or export PAs, virtually none of which have been characterized experimentally. These include multi-subunit ABC-type transporters and single-component PA-basic amino acid antiporters. The expression of *smc01652* in *S. meliloti*, annotated as encoding the substrate binding protein of a Put/agmatine ABC transporter, was induced by both agmatine and Put, consistent with the annotation [90]. PA transporters would allow intracellular PA levels to be modulated via PA import and export as well as the dispatch and receipt of PAs as chemical signals (Section 5, [37]). In addition, PA/basic amino acid antiporters might function in acid stress resistance, as described in Section 3.2.2. 

In rhizobia and other bacteria, alternative and sometimes compensatory PAs are produced under certain growth conditions. In the plant pathogen *A. tumefaciens*, only Put and Spd are normally found in cultures of wild-type strain C58. When strain C58 is grown in the presence of the Odc inhibitor difluoromethylornithine (DFMO), it produces Spm. When Spd biosynthesis is mutationally inactivated, it makes HSpd [80,91]. Growing *S. meliloti* in the presence of exogenous NSpd boosts the level of DAP and reduces levels of Spd, HSpd, and Put [60]. These collateral effects complicate the interpretation of experiments requiring the addition of PAs to cultures, for example, in the chemical complementation of PA biosynthesis mutants.

### 3.2. Functions of Polyamines in PGPR

As mentioned in Section 1, PAs affect growth, biofilm formation, and motility in many bacteria. PAs in enteric bacteria also affect the translation of mRNAs for some global transcriptional regulators and are important in DNA supercoiling [29,92]. It is not known if either of these phenomena occurs in PGPR. Some of the known effects of PAs on PGPR are summarized in Table 1 and are further described in the following sections.

#### 3.2.1. Requirement for Polyamines for Growth and Development

The requirement for endogenous PA synthesis for bacterial growth varies from dispensable to essential, depending on the species and growth conditions. For example, *Bacillus subtilis* does not require PA biosynthesis for growth [70], while the plant pathogen *A. tumefaciens* does not grow if unable to synthesize a PA having a 1,3-diaminopropane moiety such as DAP, Spd, or NSpd [80]. The majority of bacteria simply grow more slowly when PA biosynthesis genes are inactivated, or the enzymes are pharmacologically inhibited [30,93]. 

The growth of *R. leguminosarum* bv. viciae 3841 is significantly slowed in cultures that contain the irreversible Odc inhibitor DFMO. Odc activity, intracellular PA levels, and normal growth were largely restored by adding exogenous Put or Spd to the DFMO-containing cultures [82,83] (Table 1). Similarly, a *S. meliloti* Rm8530 *odc2* mutant that produced low levels of PAs and grew at 60% of the wild-type velocity grew normally if cultures were supplemented with Put or Spd [60] (Table 1). 

Some PAs inhibit the growth of diazotrophic bacteria when added to cultures. In *S. meliloti* Rm8530, exogenous 1 mM Put, Spd, or NSpd added to cultures are transported into and accumulate in the cells and modestly reduce the growth velocity of the cultures (Table 1). This may be a direct effect of increasing the intracellular level of the exogenously added PA or due to changes in the levels of other PAs in the cell that result from supplementation with exogenous PAs, as mentioned in Section 3.1. 

A recent transposon sequencing (Tn-seq) study identified the *hss* genes in *B. japonicum* USDA110 and *R. palustris* DGA009 as being essential for the growth of these strains in rich medium [94]. A PA profile for *R. palustris* DGA009 has not been reported, but other strains of the species contain HSpd as their major PA [81,86,95]. *B. japonicum* USDA110 also produces relatively high levels of HSpd [36,85]. To confirm the results of the Tn-seq analysis, growth experiments with purposely constructed *hss* null mutants should be done.

In contrast to the apparent essentiality of *hss* in *R. palustris* and *B. japonicum*, Tn-seq analyses of *S. meliloti* 1021 and *R. leguminosarum* bv. viciae 3841 showed that none of their respective PA synthesis genes were essential for growth [62,96,97]. The lack of a requirement for *hss* in *S. meliloti* is consistent with the normal growth rate of a strain 1021 *hss* mutant (Becerra-Rivera and Dunn, unpublished). This mutant might grow normally because it produces the potentially compensatory triamines Spd and NSpd [60]. In contrast, HSpd is the only triamine produced by *R. leguminosarum* bv. viciae 3841 [36]. The generation and phenotypic testing of a *hss* mutant is needed to confirm the importance of HSpd in this strain [82]. 

Under nitrogen-limited growth conditions, vegetative cells in filaments of the cyanobacterium *Anabaena* differentiate into nitrogen-fixing heterocysts. Heterocyst formation in *Anabaena* sp. PCC 7120 requires HSpd synthesis. As mentioned in Section 3.1, HSpd in *Anabaena* is synthesized by SpeA (Adc), SpeB (agmatinase), and SpeY (deoxyhypusine synthase-like Hss). Inactivation of the gene encoding any of these enzymes prevents HSpd synthesis and impairs (∆*speB* and ∆*speY* mutants) or prevents (∆*speA* mutant) diazotrophic growth in a medium with N_2_ as a nitrogen source. Nitrogen fixation does not occur in the *speA* mutant because its vegetative cells are unable to form heterocysts. The need for HSpd in heterocyst differentiation may arise from its use as a building block of the cell wall or because the lack of its production impacts nitrogen metabolism or signaling pathways necessary for heterocyst differentiation [63]. 

#### 3.2.2. Functions of Polyamines in PGPR Abiotic Stress Resistance

PGPR in the bulk soil and rhizosphere may encounter salinity, drought, pH, temperature, or oxidative stress. These stresses can significantly reduce populations of PGPR and decrease their ability to interact with plants despite a number of stress defense strategies present in bacteria. The involvement of PAs in resistance to these stresses is described in the following sections, while general stress resistance mechanisms were reviewed by [98,99,100,101,102,103,104,105,106]. For most PGPR, studies on stress resistance have dealt only with how they confer stress tolerance to a plant partner. In rhizobia, stress resistance has been extensively studied ex planta, and the following sections deal exclusively with this work.

##### Oxidative Stress

Reactive oxygen species (ROS) such as hydrogen peroxide (H_2_O_2_), hydroxyl radical (·OH), and superoxide anion (·O_2_^−^) are produced from oxygen during normal aerobic metabolism and at higher levels under abiotic stress [107]. In addition to their ability to damage macromolecules, increase mutation rates, and slow growth, ROS also have important signaling and functional roles in the rhizobia-legume symbiosis. Signaling by H_2_O_2_ and general mechanisms by which rhizobia contend with oxidative stress have been reviewed [39,104,108,109]. Like other bacteria, rhizobia fight oxidative stress mostly with enzymatic defenses, including superoxide dismutases, peroxidases, and catalases [110,111,112]. PAs represent one of the non-enzymatic defenses against ROS in bacteria [32,113]. 

TolC encodes a bacterial membrane protein required for exporting toxic substances like antibiotics, disinfectants, and eukaryotic defense compounds out of the cell. The inactivation of *tolC* in *S. meliloti* appears to increase the levels of oxidative stress based on the higher expression of genes involved in enzymatic and non-enzymatic oxidative stress resistance mechanisms. Genes whose expression is also induced in the *tolC* mutant include those encoding Orn decarboxylase *odc1* and the Arg/Lys/Orn decarboxylase *sma0682* [114]. This indicates that the *tolC* mutant may be synthesizing additional quantities of PAs in response to oxidative stress. In comparison to the *S. meliloti* wild-type Rm8530, an *odc2* mutant that produces low levels of PAs grows more slowly than the wild type and reaches less than half its cell density when grown in a minimal medium containing 0.33 mM H_2_O_2_. However, we hypothesized that the greater peroxide sensitivity of the mutant was at least partly due to its inability to make wild type levels of EPS [115], which protects *S. meliloti* from stress caused by exogenous H_2_O_2_ [116]. A metabolome analysis of the oxidative stress response caused by treating *R. leguminosarum* bv. viciae 3841 with the synthetic auxin 2,4-dichlorophenoxyacetic acid showed an increase in Cad and Put levels [117], indicating a possible direct role of these PAs in oxidative stress resistance.

The oxidative stress response in archaea is similar to that of bacteria with respect to antioxidant enzymes [118,119], but the involvement of PAs in resisting oxidative stress is not reported. 

##### Osmotic Stress

About 20% of the earth’s arable land is prone to salinity problems. High concentrations of salts cause ionic osmotic stress, while elevated levels of non-ionic solutes like sugars cause non-ionic osmotic stress. PGPR in soil may be exposed to either or both kinds of osmotic stress, especially in the rhizosphere or during plant colonization or infection [102]. PGPR show a wide range of hyperosmotic stress tolerances, and PAs participate in this by acting as chaperones to physically protect macromolecules from damage or possibly as a signal to increase the intracellular concentration of compatible solutes. The ability of PAs themselves to act as compatible solutes is controversial, mainly because they are present in much lower amounts than the well-established compatible solutes like proline, gamma-aminobutyric acid (GABA), trehalose, or sucrose [106,120,121,122].

Growth under saline conditions has opposing effects on intracellular PA levels in *Rhizobium tropici* CIAT899 and *S. meliloti* Rm8530, although both species are similarly salt tolerant. In the *R. tropici* wild type, HSpd levels and Odc activity increased a few-fold under salt stress. A CIAT899 *hss* mutant that lacked HSpd grew slower than the wild type under salt stress (0.2 M NaCl in rich medium). The growth of the mutant was partially restored when 0.1 mM HSpd was added to the cultures. These results indicate that HSpd is important for osmotic stress resistance in *R. tropici* [123]. 

Growing *S. meliloti* Rm8530 in a minimal medium with 0.3 M NaCl did not alter the Spd concentration but significantly reduced Put and HSpd levels and *odc2* gene transcription [60]. Salt shock in *E. coli* causes a rapid uptake of potassium ions and an increase in endogenous glutamate synthesis. Put, but not Spd, is rapidly exported from *E. coli* cells to prevent an overload of cations in the cytoplasm [124]. In salt-stressed *S. meliloti*, Put and HSpd export, along with *odc2* downregulation, may account for the low intracellular levels of these PAs. In *S. meliloti* 1021, salt stress decreased the expression of *odc2* and the genes for the PotD and PotF substrate binding proteins of Put and Spd-preferential ABC transporters, respectively [125]. The homologous transporters in *E. coli* function in Put and Spd import, respectively, rather than in their export. These results suggest that less endogenous PA synthesis and import from the environment lead to the low levels of Put and HSpd found in salt-stressed *S. meliloti* [60,125]. Furthermore, an *S. meliloti* Rm8530 *hss* mutant grew like wild type under salt stress, while a *cansdh* grew more slowly. Exogenous Spd restored the growth of the *cansdh* mutant under salt stress, indicating that Spd is important in salt stress resistance in *S. meliloti* (M. Dunn, unpublished), perhaps by acting as a molecular chaperone. In contrast, the levels of HSpd and Put may be decreased to achieve cytoplasmic ionic balancing. 

As in *S. meliloti* [60,125], the expression of the *R. etli* CE3 *odc2* homolog is lower under salt stress [126]. Changes in PA levels in salt-stressed *R. etli* have not been reported to see if they correlate with the changes in *odc2* gene transcription. 

The acid- and salt-tolerant *S. fredii* strain P220 accumulates intracellular glutamate and potassium and loses HSpd under salt stress. The failure to detect HSpd in supernatants of the cultures indicates that it was not exported from the cells [127], but we calculate that HSpd in the culture supernatants, if analyzed directly without concentration, would likely be undetectable. It was proposed that the reduction in HSpd levels might aid in cytoplasmic ionic charge balancing under osmotic stress [127]. 

In contrast to most rhizobia, PAs levels in the cyanobacterium *Synechocystis* sp. PCC 6803 increased under salt or osmotic stress [128], where it was proposed that they act as compatible solutes [121]. Adc mRNA in this organism was more stable under salt stress, although total Adc activity was unchanged between control and high salt conditions [128]. 

##### Acid Stress

Acid soils, defined as those with a pH of 5.5 or less, comprise as much as 40% of arable lands worldwide. In many rhizobia-legume combinations, both nodulation and nitrogen fixation are reduced in acid soils [100]. Rhizobia are also exposed to acidic conditions in plant rhizospheres and the symbiosome space surrounding bacteroids in root nodules [102], and so these bacteria have multiple acid resistance mechanisms [100,101,129].

The HSpd that accumulates in acid-stressed *S. fredii* P220 was proposed to act as a chaperone to protect macromolecules from acid-induced degradation [127]. PA synthesis also consumes protons and alkalinizes the cytoplasm. Bacteria such as *E. coli* have acid-inducible isozymes of Adc, Odc, and Ldc paired with cognate basic amino acid-PA antiporters. The basic amino acid decarboxylation reactions consume protons and thus raise the intracellular pH in cells under acid stress. The PA produced in the decarboxylation reaction is exported from the cell by the antiporter in exchange for a molecule of the decarboxylase’s basic amino acid substrate, thus constituting a cycle of decarboxylation, product export, and substrate import [130]. One region of the *S. meliloti* symbiotic plasmid (pSyma) encodes a putative Put/Orn antiporter next to the *odc1* gene and located close by are a possible agmatine/Arg antiporter and putative Adc [36]. *S. meliloti* encodes two other possible basic amino acid/PA antiporters in other regions of pSyma, one of which was shown to be essential for symbiosis by Tn-seq analysis [62]. The function of the decarboxylase and antiporter genes on the symbiotic plasmid suggests a role in symbiosis, most logically in acid stress resistance. Nevertheless, an *S. meliloti odc1* mutant grew like wild type at pH 5.5, suggesting that Odc1 is not required for resistance to this relatively mild acidity [60]. Further testing of this mutant and of the other pSyma genes potentially involved in acid resistance is being done in our laboratory. 

The expression of a Spd/Put ABC transporter was reduced under acid growth conditions in *Rhizobium freirei* PRF 81, which belongs to the “*R. tropici* group” of bean nodulating rhizobia [129]. In *S. meliloti* 1021, *potD* encodes the substrate binding protein of a putative TRAP-type Put transporter and was progressively and permanently downregulated over the course of short-term growth at low pH [131]. In the case of acid-stressed *S. meliloti*, decreased import of external Put from the environment could prevent its interfering with endogenous Put biosynthesis and the proton consumption that is accomplished as part of the acid resistance mechanism described above. However, the *R. freirei* PRF81 genome sequence does not appear to contain basic amino acid/PA antiporters nor *odc* homologs of those on the *S. meliloti* pSym.

The growth of *S. meliloti* Rm8530 at pH 5.5 caused a several-fold decrease in the level of Put, while the amounts of Spd and HSpd were unaffected. An *odc2* mutant that made very low levels of Put, HSpd, and Spd grew much more slowly than the wild type at pH 5.5. Growth of the mutant at pH 5.5 was completely restored by exogenous Spd and partially restored by Put or NSpd (HSpd was not tested due to lack of a commercially available product) [60]. These results suggest that Put (itself or as a precursor of Spd), NSpd, and Spd are able to lessen the impact of acid stress on growth and may be functionally interchangeable.

In *S. meliloti* 2011, grown at different pHs in chemostat cultures, intracellular Put levels increased at pHs above (7.4) or below (6.1) neutrality. However, transcriptomic and proteomic analyses did not show changes in genes for PA biosynthesis or transport [132]. The acid-induced decrease in Put in *S. meliloti* Rm8530 batch cultures and its increase in strain 2011 chemostat cultures are probably due to differences in the experimental approaches and the fact that strains 2011 and 1021 (the latter is the parent strain of Rm8530) do show important phenotypic differences despite being derived from *S. meliloti* strain SU47 [133].

##### Temperature Stress

Global warming is increasing heat stress on PGPR and plants. High temperatures cause protein misfolding and inactivation [98,121]. Heat shock in *R. etli* CE3 increased the expression of genes encoding chaperones and proteases that aid in protein folding and in degrading misfolded proteins, respectively. These responses do not occur in *R. etli* exposed to salt shock [126]. 

Little has been reported on PAs as related to heat stress resistance in PGPR. Growing *S. meliloti* Rm8530 at 37 °C rather than its “optimal” temperature of 30 °C significantly increased both its growth rate and its intracellular content of Spd. A Rm8530 *odc2* unable to synthesize normal levels of Spd, Put, or HSpd grew at about half the rate of the wild type at 37 °C, while the mutant genetically complemented with the *odc2* gene grew like wild type. This suggests that Spd or other PAs are important for the growth of *S. meliloti* under heat stress (V. Becerra-Rivera, unpublished).

Cold stress creates problems for bacteria by increasing membrane rigidity and decreasing the rate of enzyme reactions [98,121]. The general response of *Mesorhizobium* sp. strain N33 to cold shock was to adjust gene expression to achieve growth reduction or cessation. In addition, strain N33 under cold stress upregulated expression of the gene for the ATPase subunit of an ABC-type PA transporter, possibly indicating increased transport of PAs as protective agents [134]. 

#### 3.2.3. The Importance of Polyamines in Motility and Biofilm Formation

PAs present in the environment markedly affect chemotaxis, EPS production, biofilm formation, and motility in some PGPR [36,38]. Chemotaxis to root exudates is an essential first step for the establishment of PGPR–plant interactions: Bacteria that arrive first at the root will have the widest choice of nutrients and hence a growth advantage [24]. Put, Cad, and Spd are chemoattractants for *Pseudomonas putida* KT2440, which can utilize these PAs as carbon or nitrogen sources for growth in cultures. All 3 of these PAs bind to the McpU ligand binding domain, while Spm, Orn, GABA, or proteinogenic amino acids do not. A *P. putida* KT2440 *mcpU* mutant was severely compromised in its ability to colonize corn roots [44]. The McpU homolog in *S. meliloti* allows chemotaxis towards proline, which is found in alfalfa seed exudates, but it is not known if it is also a chemoreceptor for PAs [135]. 

The second messenger bis-(3′-5′) cyclic diguanosine monophosphate (c-di-GMP) regulates motility and biofilm formation in many bacteria. In *S. meliloti*, the PA signal detection/transduction system comprised of the NspS and MbaA proteins affects biofilm, EPS production, and motility in response to specific exogenous PAs. The NspS-MbaA-dependent phenotypic responses to PAs differ depending on whether the quorum sensing (QS) system is functional. This is consistent with the multiple interacting regulatory circuits, including QS, that regulate biofilm formation, EPS production and motility in *S. meliloti*. We speculate that the NspS-MbaA system modulates these symbiotically important phenotypes in *S. meliloti* in response to host plant-produced PAs [136].

Swimming motility in an *S. meliloti* Rm8530 *hss* null mutant was reduced by one-third relative to the wild type, while a *cansdh* mutant was unaffected in swimming. Swimming by a *hss cansdh* double mutant was reduced by nearly 60% compared to the wild type (V. Becerra-Rivera and M. Dunn, unpublished results). The more pronounced phenotype of the double mutant probably results from its inability to produce either of the partially compensatory triamines Spd and HSpd. 

Inactivation of the *hss* gene in *R. etli* CNPAF512 abolished its ability to swarm, indicating that HSpd is essential for this type of motility [137]. Like *R. etli* strain CFN42 [36], strain CNPAF512 probably lacks the ability to produce Spd which might compensate for the lack of HSpd in the mutant. 

## 4. A Brief Review of the Roles of Polyamines in Plants

While higher plants have both the Odc and Adc pathways for Put production [33,138], Adc is usually the sole or more important pathway [139]. Put is a precursor of Spd and Spm in all plants and of TSpm in some plants. These PAs are synthesized by Spd, Spm, and TSpm synthases, respectively, as described in Section 3.1 (Figure 2). The triamine NSpd is found in non-vascular plants and some higher plants [140,141,142]. The production of Cad by Ldc is restricted to *Leguminosae* and *Solanaceae* [143]. 

Broadly generalizing, Spd is required in plants for growth, and Spm and Put are important in stress resistance [139]. Manipulating the PA content in plants by exogenous application or metabolic engineering has shown that PAs are involved in embryogenesis, fruit development and maturation, flower and chloroplast development, organogenesis, and senescence and stress responses. TSpm, for example, controls stem elongation in *A. thaliana* by promoting the expression of upstream open reading frames (uORFs) that control translation of genes involved in xylem differentiation, which affects stem elongation [79,144]. In fact, some genes required for plant PA biosynthesis are regulated by uORFs, probably in response to specific PAs [145]. Spm and its degradation products trigger plant defense reactions to pathogens (for reviews, see [39,40,146,147].

In legumes, L/Odcs are the starting points for the production of Cad and HSpd for making alkaloids that provide chemical defense against herbivorous insects [143,148,149]. In the genus *Crotalaria* (“rattlepods”, Fabaceae), HSpd is incorporated into the structure of the pyrrolizidine alkaloid monocrotaline. Plant-encoded Hss produces HSpd for monocrotaline synthesis, although the plants need to be nodulated by *Bradyrhizobium* spp. to produce the alkaloid. The restriction of alkaloid synthesis to nodulated plants might ensure their sufficiently high nitrogen status for alkaloid production [149]. 

PAs protect plants from abiotic stresses by acting as molecular chaperones to protect macromolecules from physical and chemical damage. PAs in plants also stabilize plasma membranes and significantly affect the activity of various ion transporters, both of which are important for growth and stress resistance [150,151]. A recent study found that treatments affecting the formation of conjugated forms of Spd and Spm correlated with drought resistance, root plasma membrane stability, and higher H^+^-ATPase activity [152]. Pas also interact with phytohormones to promote stress resistance (Section 5.2.2).

PA catabolism regulates plant stress response pathways by producing nitric oxide (NO), H_2_O_2_, and GABA, all of which also have important signaling functions in plant–microbe interactions [41]. These signals arise as products of PA catabolism by diamine and PA oxidases [153]. Plant defense responses triggered by different biotic and abiotic stresses increase ROS levels. These ROS also serve as chemical signals during the interaction of legumes and rhizobia and are required for the development of an efficient symbiosis [99,102]. PAs are also effective scavengers of ROS, increase the activity of antioxidant enzymes, and trigger ROS-defensive signaling pathways (for reviews, see [105,122,146,154,155]). Thus, in addition to being the agents of oxidative stress and associated damage, ROS in plants act as key physiological signaling and regulatory molecules. In terms of PA metabolism, a balance must be struck between the protective effects of PAs against ROS and ROS production by PA catabolism [104,146,156].

Analyses of abiotic stress-tolerant legume genotypes have revealed changes in their PA content in comparison to less tolerant lines. For example, a drought-tolerant line of *Lupinus luteus* (yellow lupin) had a high Spd level and was characterized by higher plant and seed biomass than non-tolerant lines under drought stress [157]. Salt-stress tolerant genotypes of *M. truncatula* [158] and *G. max* [159] had high levels of Spm and Put, and Spd and Spm, respectively, when grown under salt stress. In both cases, PA catabolism by PAO was diminished, and in *G. max*, antioxidant enzyme activities increased [159]. Because many different kinds of abiotic stress ultimately trigger oxidative stress, the ability of PAs to increase a plant’s resistance to diverse stress conditions is, in part, a measure of their ability to resist oxidative stress. 

PAs also aid in abiotic stress resistance in plants by increasing the production of GABA, which moderates the negative effects of abiotic stress by affecting osmoregulation, antioxidant defense, and molecular signaling [160,161]. GABA synthesis in plants results from the degradation of PAs by diamine oxidase (DAO, EC 1.4.3.6) and by the action of glutamate decarboxylase (GDC, EC 4.1.1.15). Salt stress imposed on soybean seedlings caused a rapid increase in DAO activity and *dao* gene transcription, and GABA content, with a concomitant decrease in Put and Spd levels. The inhibition of DAO with the specific inhibitor aminoguanidine reduced seedling GABA content by nearly 40%, suggesting that this proportion of total GABA is produced by DAO [162]. Fang and co-workers [163] showed that exogenous Spd alleviated the inhibition of soybean seedling growth caused by salt stress. Spd treatment was accompanied by an increase in GABA content and a reduction in malondialdehyde (an indicator of membrane oxidative damage) and H_2_O_2_ levels, concomitant with an increase in ROS-detoxifying enzyme activities [163]. Similarly, exogenous Put prevented the growth inhibition of soybean seedlings under salt stress and increased superoxide dismutase, catalase, and peroxidase activities to near the levels found in unstressed plants. Exogenous Put reduced the levels of H_2_O_2_ and superoxide as well as that of malondialdehyde [164]. 

## 5. Polyamines in PGPR–Plant Interactions

We have seen that endogenous PAs or those taken from the environment can affect diazotrophs and plants living separately, but what effects do they have during the interaction of plants with PGPR? The fact that PAs modulate the interaction of plant pathogens with their hosts is well documented, to the extent that phytopathogens are known to alter plant PA metabolism to their own benefit [39,165,166,167,168]. The roles of PAs in mutualistic plant–PGPR interactions are much less clear, and most work done in the area has been with rhizobia and legumes [37,39,105]. In this section, we examine how PA exchange between symbiotic partners affects developmental processes and stress resistance in both plants and PGPR.

### 5.1. Polyamines Found in Root Nodules

The free PAs commonly found in legume root nodules include Put, HSpd, Spd, Cad, and Spm. Put in nodules is synthesized by both symbionts while Spd, Cad, and Spm are usually made by the plant partner [36,37,120]. In contrast, HSpd in nodules is contributed by the microsymbiont, although some legumes are able to synthesize it using a deoxyhypusine synthase-like Hss that transfers an aminobutyl group from Spd to Put to produce HSpd [33,140,143,169].

In nodules formed by *R. legminosarum* bv. viciae 3841 on pea mutants unable to support nitrogen fixation, plant *odc* gene expression was highly induced in the ineffective nodules relative to the effective nodules formed on the wild-type pea line. The transcriptional expression of *adc* was similar in both effective and ineffective nodules. The higher *odc* expression in ineffective nodules indicates that it may function as part of a defense response against the formation of non-functional nodules, while the Adc serves for the housekeeping synthesis of Put [170] (Section 4).

The uncommon PA 4-aminobutylcadaverine (AbCad) is found in nodules of adzuki bean and common bean formed by *B. japonicum* A1017 and *R. tropici* CIAT899, respectively. The synthesis of AbCad in nodules on these species depends on their containing high levels of Cad, to which the bacteroid Hss adds an aminobutyl group from Put to form AbCad [82,123,171,172]. Bacteroids of an *R. tropici* CIAT899 *hss* mutant failed to produce HSpd and AbCad in bean nodules, while the wild type made high amounts of each [123]. These findings confirm the bacterial origin of HSpd in bean nodules and the role of Hss in producing AbCad.

Bacteroids in nodules are notable for their high HSpd content [123,172,173,174]. Nodules from soybean, faba bean, and kidney bean contain roughly equal amounts of HSpd and Spd, while those of cowpea, mung bean, siratro, and *Sesbania* spp. have high HSpd to Spd ratios. The fact that different PA patterns occur in nodules of different species of plants able to host the same *Rhizobium* or *Bradyrhizobium* species indicates that the PA content in nodules is mostly controlled by the plant host and not the microsymbiont [174]. It would be interesting to further explore this using a host plant that is nodulated by rhizobia that differ in the types of PAs they produce. 

Unlike legumes, the non-legume shrub *Alnus* spp. has extremely low quantities of PAs in its nodules [174]. Nodules from different *Alnus* species contained Spd and occasionally Cad and Put but lacked HSpd [84,174]. The low levels of PAs in these nodules may relate to their high content of lignified tissue, which in senescent nodules correlates with a low PA content [84,174].

A *R. leguminosarum* bv. viciae strain was modified by replacing its native symbiotic plasmid with that from an *R. leguminosarum* bv. phaseoli strain, thus allowing it to form indeterminate nodules on pea while the native bv. viciae strain formed determinate nodules on bean. Transcriptome comparisons of bacteroids from both hosts showed that genes for Put transporter components, a L/Odc and Hss were upregulated in bacteroids isolated from determinate nodules but not in bacteroids from indeterminate nodules [175]. The upregulation of the Put transporter genes suggests the presence of sufficient quantities of Put around the bacteroids in determinate nodules to induce the transporter [90], while upregulation of the two genes for PA biosynthetic enzymes suggests that more PAs are produced by the bacteroids in these nodules than in the indeterminate ones [125].

Another example of host-dependent expression of PA biosynthesis genes was found by proteome analysis of *R. leguminosarum* bv. viciae UPM791 bacteroids isolated from indeterminate nodules formed on pea and lentil. Those from pea showed much higher expression of putative Ldc, Odc, and DABA AT enzymes in pea relative to lentil [176]. Bacteroids of strain UPM791 from both pea and lentil contained Put, NSpd, Spd, and HSpd, while cells grown in culture contained only HSpd and Put. Unexpectedly, the deletion of gene encoding DABA AT did not alter the PA profile of bacteroids from either host or of cells grown in culture [177].

Nodules on the non-legume *Parasponia andersonii* formed by *Bradyrhizobium* spp. CP-283 contained similar levels and types of PAs as found in most legume nodules, predominantly Put and HSpd, with much lower levels of Spd, Cad, and Spm [174]. A comparison of the PAs in stem and root nodules on the non-legume *Sesbania rostrata* inoculated with *A. caulinodans* ORS571 showed they were similar in both types of nodules [174], despite their different morphologies and the much higher level of nitrogen fixation and metabolic activity in stem nodules [178].

### 5.2. Involvement of PAs in PGPR–Plant Interactions

#### 5.2.1. Effects of PAs on Growth, Differentiation, and Metabolism of the Symbiotic Partner

Plants treated with exogenous PAs generally show increased growth, accelerated development, higher photosynthetic efficiency, and greater stress resistance [40,153,170,179]. Examples of how the manipulation of PA levels in plants affects their interaction with PGPR are summarized in Table 1, and some examples are further described here. 

Put added to the nutrient solution of chickpea or vetch inoculated with indigenous or commercial inoculum strains of *R. leguminosarum* significantly increased plant nodule and shoot biomass, chlorophyll, and total nitrogen [180]. In the *R. galegae*-goat’s rue symbiosis, 10 or 50 µM Put included in the agar growth medium increased nodule numbers and biomass per plant, and nitrogen-fixing activity [181], possibly by increasing acid stress resistance and the root-binding ability of *R. galegae* [182]. Treatment of the *G. orientalis* plants with 100 µM Put or Spd reduced nodule numbers and nodule biomass per plant to below the levels of the untreated control. Spm at this concentration had little effect [181]. It was hypothesized that the reduction in nodulation caused by 100 µM Put was due to this concentration being inhibitory for the growth and root attachment ability of *R. galegae* [181]. 

Exogenous Put (0.1 mM) increased the number of nodules formed and acetylene reduction activity in pea inoculated with *R. leguminosarum* bv. viciae 3841, while 1 mM Put significantly decreased nitrogenase activity but not nodulation. Wild-type plants treated with 5 mM Put were very sparsely nodulated and fixed no nitrogen. However, nodulation and nitrogen fixation in transgenic plants with greatly reduced levels of DAO activity were not inhibited by 5 mM Put. Possibly, the high Put levels inhibit bacteroid growth or development via an oxidative burst from its catabolism by DAO. This would not occur at such high levels in transgenic plants with lower DAO activities [183].

Cucumber inoculation with *A. baldaniorum* Sp 245 (formerly *A. brasilense* Sp 245) increased Put and H_2_O_2_ levels and DAO activity and lowered the content of cell wall phenolics. The higher Put content may allow DAO to produce more H_2_O_2_ and other ROS that, along with the reduced concentrations of cell wall phenolics, promote growth of the roots [184]. In wheat, inoculation with *A. baldaniorum* Sp 245 reduced root superoxide levels by nearly one-third [185].

PAs may also control the growth of some diazotrophs during their interaction with plants. Species of *Frankia* in free life fix nitrogen in vesicles, which develop from the hyphae. Hyphal growth and integrity and vesicle formation were significantly inhibited by exogenous Spd or Spm used as a nitrogen source by *Frankia* grown in cultures [84]. 

In *B. japonicum*, the regrowth of bacteroids isolated from soybean nodules was inhibited by micromolar concentrations of Spd or Spm [186]. While Spm is absent or present at very low levels in soybean nodules, Spd accounts for about one-third of their total PA content [174,187], indicative of a concentration sufficient to retard bacteroid growth.

PAs likely participate in controlling nodule formation in the *B. japonicum*/soybean interaction apart from their direct effects on bacteroid growth. The leaves, roots, and nodules of the supernodulating soybean mutant En6500 had a strikingly lower Spd/Put ratio than the parental cv. Enrei. This suggested that there was less conversion of Put to Spd in the supernodulating line. Foliar application of the SAMdc inhibitor MDL74038 significantly reduced Spd levels in the parental cv. Enrei and doubled the number of nodules formed. Consistent with this, the foliar application of Spd or Spm drastically reduced the number of nodules formed by the supernodulating En6500 mutant. This suggests that the total amount or ratio of different PAs modulate the number of nodules formed on soybean, in conjunction with phytohormones called brassinosteroids. The hypothesis from this work is that low levels of brassinosteroids in the supernodulating line suppress Spd synthesis and lead to excessive nodulation [187,188]. Our current understanding of autoregulation of nodulation is incomplete, and the process is controlled by a great number of plant signal receptors that respond to nitrogen availability, phytohormones, and chemical signals generated during the symbiosis. While PAs have only been indirectly implicated in autoregulation of nodule numbers in soybean (see above), they could also conceivably participate in modulating autoregulation control points like mRNA stability and translation efficiency [189].

Gene mutagenesis and Tn-Seq experiments showed that two arginase homologs and the Orn decarboxylase homologs Odc1 and Odc2 were required for an efficient symbiosis between *S. meliloti* and *M. truncatula* [62]. The inactivation of Odc2 in *S. meliloti* Rm8530 also decreases its symbiotic efficiency on *M. sativa* [115]. The Tn-Seq experiments also showed that the *S. meliloti* genes encoding two Arg transporters, four Put ABC transporters, Hss and DABA AT were required for an efficient symbiosis with *M. truncatula* [36,62]. The symbiotic requirement for Hss and DABA AT indicates a role for endogenously produced HSpd and NSpd in bacteroids, and the increased transporter activity might serve to import plant-produced PAs. Tn-Seq analysis of *R. leguminosarum* bv. viciae in symbiosis with pea plant revealed that the *hss* gene was essential in both undifferentiated and bacteroid forms of the microsymbiont isolated from nodules but dispensable for growth in the rhizosphere or for root colonization [190]. 

The PGPR *B. subtilis* OKB105 increased tobacco plant growth by exporting endogenously synthesized Spd, which increased the expression of genes for expansions and decreased expression of the *ACO1* gene for ethylene biosynthesis. Lower levels of ethylene favor plant growth. Strain OKB105 null mutants in *speB* (agmatinase) or a transporter required for Spd export (*yecA*) failed to stimulate plant growth. Exogenous Spd or genetically complemented mutants were effective in promoting plant growth [191]. *Bacillus megaterium* BOFC15 secretes Spd in vitro and, when inoculated onto *Arabidopsis,* increased plant Spd, Spm and abscisic acid (ABA) levels and drought stress resistance. The hypothesis from these results is that Spd secreted by *B. megaterium* caused an increase in ABA levels in the plant, and this imparts higher drought tolerance [192].

A PA ABC transporter is required for the normal development of nitrogen-fixing heterocysts in *Anabaena* sp. PCC 7120. The transporter may be required to maintain PA homeostasis or for transferring HSpd, which is required for heterocyst development, from vegetative cells to the heterocysts [63]. 

#### 5.2.2. Polyamines in PGPR–Plant Associations under Abiotic Stress

The ability of PGPR to enhance the abiotic stress resistance of plants has been amply demonstrated and was recently reviewed [18,27,154]. In general, PGPR potentiate host responses that allow them to better tolerate abiotic and biotic stresses and promote better growth in general, which itself increases the stress-fighting ability of plants.

The PGPR *Stenotrophomonas rhizophila* DSM14405 grown in the presence of oilseed rape root exudates had several-fold higher transcription of the *mdtI* and *mdtJ* genes, which encode a Spd transport system [193]. In *E. coli*, MdtIJ activity is required to prevent Spd toxicity in mutants unable to acetylate and thus inactivate “excess” endogenous Spd [194]. Spd export from *S. rhizophila* via MdtIJ was not experimentally demonstrated but was hypothesized to be involved in this PGPR’s ability to increase plant growth and stress resistance [193]. *S. rhizophila* DSM14405 co-inoculated with *B. japonicum* onto soybean increases the symbiotic effectiveness of the nitrogen-fixing interaction under salt stress. The export of Spd or glucosylglycerol (an osmoprotectant) by DSM14405 is a possible mechanism for the synergistic effects of co-inoculation with the two species [195].

The experimentally supported metabolic model proposed by Christen and co-workers asserts that Arg and succinate produced by *M. truncatula* are actively transported into *S. meliloti* bacteroids and co-metabolized to generate energy and reduce power for nitrogen fixation [62]. The catabolism of Arg to Put by the Odc and Adc/agmatine routes releases ammonium and consumes protons, which is hypothesized to protect the bacteroids against acid conditions normally present in the symbiosomes [62,99,102]. As mentioned above (Section 3.2.2 and Section 5.2.1), and consistent with this role for PAs in acid stress resistance, gene mutagenesis, and Tn-Seq experiments showed that two arginase homologs and the Orn decarboxylase homologs *odc1* and *odc2* were required for an efficient symbiosis between *S. meliloti* and *M. truncatula*. [62]. 

In *Galega orientalis* [181] and *M. truncatula* [196], treatments with exogenous PAs increased nodulation and nitrogen fixation, as well as the biomass of shoots, roots, and nodules under normal conditions and salt stress. The conclusion from these studies is that changes in the physiology or metabolism of the nodule are not caused by PAs at that site but result from secondary signals generated by PA action in the shoots. PAs have been shown to have regulatory interactions with hormones like ethylene, ABA, salicylic acid, and brassinosteroids [26,161,196,197]. Exogenous Spd or Spm in the nutrient solution supplied to *M. truncatula* inoculated with *S. meliloti* 1021 increased the expression of brassinosteroid synthesis genes and increased symbiotic efficiency under salt stress [196]. Treatment of salt-stressed *M. sativa* inoculated with *S. meliloti* GR4 without or with ABA pre-treatment increased Spd and Spm levels in nodules, increased antioxidant enzyme activities, and improved their nitrogen fixation capacity [198]. 

The positive effect of PAs on both plant and PGPR abiotic stress resistance is one reason for the generally beneficial effects of PGPR on plant growth. In addition, the development of *Rhizobium*-legume interactions may require signaling pathways or other metabolic changes that result from responses to different abiotic stresses that occur during the normal course of symbiosis. Both the micro- and macrosymbiont participate in these stress-induced responses [102], some of which are known to be modulated by PAs.

Bacteroids of wild type *R. tropici* CIAT899 isolated from bean plants grown under salt stress had higher levels of HSpd than bacteroids from unstressed plants. A CIAT899 *hss* null mutant grew more slowly than the wild type under salt stress in vitro. Plants inoculated with the *hss* mutant had significantly lower nodule fresh weights than those of plants inoculated with the wild type under both control and salt stress conditions [123]. The beneficial effect of the *R. tropici hss* on nodule mass and salt resistance agree with their importance in the symbioses of *S. meliloti* and *R. leguminosarum* with their respective hosts described in Section 5.2.1.

In the *A. brasilense* commercial inoculant strains Cd and Az39, Cad exported from the cells was implicated in plant growth promotion and increased osmotic stress resistance [75,76]. Exogenous Cad applied to rice seedlings at a concentration similar to that found in *A. brasilense* culture supernatants promoted root and shoot growth, similar to the effect of treating seedlings with *A. brasilense* Az39. The growth-promoting effects of exogenous Cad and inoculation with strain Az39 also occurs in plants under osmotic stress [75]. However, for other strains of *A. brasilense* inoculated onto wheat or *Arabidopsis*, auxin production by the bacteria and inhibition of plant ethylene biosynthesis appears to be responsible for plant growth promotion [199,200].

## 6. Conclusions and Perspectives

In PGPRs and plants, endogenous PA biosynthesis and the uptake of PAs from the environment are important in determining intracellular PA levels and, accordingly, their physiological effects [201]. A summary of the general effects of PAs on plant–PGPR interactions is shown in Figure 3, although it is conceptually and experimentally difficult to separate the effects of PAs on symbiotic partners separately versus effects that occur only during their interaction. PAs excreted into the environment can also serve as signal molecules in PGPR–plant interactions, although this aspect has not been explored in great depth. The possible or proven mechanisms by which PAs exported by PGPR or nodule-forming rhizobia affect plant growth and stress resistance include stabilizing plant plasma membranes, affecting the activity of plant ion transporters, decreasing oxidative damage to cells by increasing antioxidant defenses or by interacting with phytohormones to promote stress resistance (Section 4). PAs made by legumes also participate in limiting bacteroid proliferation and regulation of nodule numbers (Section 5.2.1). In PGPR, the perception of environmental PAs is important in promoting chemotaxis, while both endogenous and exogenous PAs affect motility, biofilm formation, and EPS production (Section 3.2.3). Because PAs in either plant or PGPR may act as signals, their ability to trigger physiological or morphological changes or increase stress resistance may not require remarkable changes in their levels, a potentially misplaced criterium for their importance in many studies. 

PA biosynthesis or availability of PGPR in soil or in their interaction with plants are important as modulators of stress resistance. In soil, the ability to survive for long periods is important in maintaining PGPR populations and giving competitive advantages in the rhizosphere to specific PGPR species [202], and PAs have an important role in this. 

The efficient design of effective bioinoculants containing synthetic communities of PGPR requires detailed knowledge of the physiological and genetic traits of the organisms and their interaction with one another and the environment, including different plant hosts, and how to translate laboratory studies done on a few model organisms to a broader range of plants in the field [203]. Could manipulating the kinds or quantities of PAs exported by PGPR enhance their ability to increase the growth of specific crops? As described in Section 3.1, PGPR produce a range of different PAs, although there is considerable overlap in the PA profiles within some species and even in some genera and classes (Section 3.1). It is not known if differences in the types of PAs produced by PGPR function as determinants of host specificity: This would depend largely on the Pas excreted by the PGPR and how they are perceived by different plants. We suspect that Pas produced by PGPR are not major determinants of host specificity since many plants interact promiscuously with PGPR or rhizobia that produce different types of Pas, while specific wide host range PGPR can interact with many different plant hosts [43,50,204]. 

**Table 1 plants-12-02671-t001:** Effects of polyamines on PGPR in free life or in association with plants.

PGPR	Experimental System or Treatment	Results	Interpretation	References
*Anabaena* sp.	Diazotrophic growth of *speA* (Adc), *speB* (agmatinase), or *speY* (*hss* synthase homolog) null mutants	Reduced diazotrophic growth. *speA* mutant cannot form heterocysts	HSpd is needed for heterocyst differentiation as a cell wall component, or its absence affects nitrogen metabolism and/or signaling processes	[63]
*Azospirillum* *baldaniorum* sp. 245	Inoculated on cucumber	Inoculation correlated with increased Put and H_2_O_2_ levels, diamine oxidase (DAO) activity, and reduced cell wall phenolics content	Higher Put content allows DAO to make more H_2_O_2_ and other reactive oxygen species (ROS). This and the lower concentration of phenolics promote root growth	[184]
*Azospirillum* *brasilense* Az39	Rice inoculated with strain Az39 or treated with exogenous Cad, with or without osmotic stress (mannitol)	Significantly increased osmotic stress resistance and abscisic acid (ABA) production for both treatments	Cad excreted by strain Az39 promoted growth and osmotic stress resistance in rice	[75]
*Bacillus megaterium* BOFC15	Inoculated on *Arabidopsis*	BOFC15 secretes Spd in vitro. When inoculated on *Arabidopsis,* it increased plant Spd, Spm, and ABA levels, and drought stress resistance	Spd secreted by *B. megaterium* increases plant ABA levels, thus increasing drought tolerance	[192]
*Bacillus subtilis* OKB105	Inoculated on tobacco	The strain exports Spd, causing increased expression of genes for expansins and decreased expression of the *ACO1* gene for ethylene biosynthesis. Spd biosynthesis mutants do not have this effect	Spd exported by strain OKB105 results in lower ethylene content in the plants and favors growth	[191]
*Bradyrhizobium diazoefficiens* 110*spc4*, *Rhodopseudomonas palustris* CGA009, *Rhizobium tropici* CIAT899^T^	Growth in cultures following Tn-seq mutagenesis (*B. j.*, *R. p*.) or *hss* inactivation (*R. t.*)	Few *hss* mutants recovered (Tn-seq mutagenesis), or slower growth of *hss* mutant	HSpd important for growth	[94,123]
*Bradyrhizobium japonicum* 138NR	Growth in culture with exogenous PAs	Regrowth of bacteroids isolated from soybean nodules, but not of culture-grown cells, is prevented by 0.1 mM Spd or Spm, but not by Put	Spd and Spm inhibit proliferation of bacteroids in nodules	[186]
*Bradyrhizobium japonicum* USDA 110 (now *B. diazoefficiens* USDA 110^T^ [205])	Inoculation on soybean super-nodulating and parent cultivar, without or with foliar application of Spd or Spm, or the SAMdc inhibitor MDL74038	PA treatment decreased the number of nodules formed on the super-nodulating cv. and decreased root growth. Application of MDH74038 to the parent cv. increased nodule number	PAs function as a systemic regulator of nodulation by affecting the plant rather than the microsymbiont	[187,188]
*Frankia spp.*, various strains	Cultures grown with exogenous Spd or Spm	Formation of vesicles (where nitrogenase is located) is inhibited, hyphal degradation occurs	PAs inhibit growth and vesicle formation	[84]
*Pseudomonas putida* KT2440	Inoculated on corn	Ability of a *mcpU* PA chemotaxis receptor mutant to colonize plants was greatly reduced	Chemotaxis towards PAs favors root colonization	[44]
*Rhizobium etli* CNPAF512	*hss* null mutant	Mutant unable to swarm	HSpd required for swarming motility	[137]
*Rhizobium galegae* HAMBI 540^T^	Inoculated on goat’s rue, plants grown without or with 10 or 50 µM exogenous Put or Spd	Plant acid stress resistance and binding of *R. galegae* to roots enhanced by PA treatment	The exogenous PAs increase root attachment and colonization	[181]
*Rhizobium galegae* HAMBI 540^T^	Inoculated on goat’s rue, plants grown without or with 100 µM exogenous Put, Spd or Spm	Nodule number and biomass per plant with Put or Spd treatment lower than those of untreated control. Spm had little effect	High concentration of Put or Spd reduces *R. galegae* growth and root attachment ability	[181]
*Rhizobium leguminosarum* indigenous or commercial inoculum strains	Inoculation on chickpea or vetch with or without single-dose Put treatment	Put treatment significantly increased nodule and shoot biomass, chlorophyll, and total nitrogen	Stimulation of plant growth with Put treatment not dependent on *R. leguminosarum* strain. Mechanism of growth stimulation unknown	[180]
*Rhizobium leguminosarum* bv. viciae 3841	Growth in cultures without or with 2,4-dichlorophenoxyacetic acid	2,4-dichlorophenoxyacetic acid provokes oxidative stress	Intracellular levels of Put and Cad increase as a protective response against oxidative stress	[117]
*Rhizobium leguminosarum* bv. viciae 3841	Inoculated on pea plants grown with 0, 0.1, 0.5, 1, or 5 mM exogenous Put	Exogenous Put at 0.1 mM gives a slight increase in nodule number and a 50% increase in nitrogenase activity per plant. Put at ≥1 mM decrease these parameters relative to control without Put	Endogenous Put at too high a level is inhibitory to nodule function	[183]
*Rhizobium leguminosarum* bv. viciae 3841	Inoculated on transgenic pea lines with normal or severely reduced DAO activity, under control conditions or with exogenous Put	Under control conditions, no correlation between nodule number or nitrogen fixation and DAO activity, but with exogenous Put, small increase in nodule number and big increase in nitrogen fixation.	DAO activity reduces inhibitory levels of PAs in nodules	[183]
*Rhizobium leguminosarum* bv. viciae 3841	Growth in cultures without or with the Odc inhibitor DFMO	Inhibitor reduces PA biosynthesis and growth: Exogenous Put or Spd largely overcomes inhibition	PA biosynthesis required for wild type growth	[82,83]
*Rhizobium leguminosarum* bv. Viciae with native viciae pSym or with introduced pSym from bv. Phaseoli	Strains with bv. Viciae and bv. Phaseoli pSyms form determinate and indeterminate nodules, respectively, on bean	Put transporter components, *l/odc* and *hss,* upregulated in bacteroids from determinate but not indeterminate nodules	PA transport and endogenous synthesis higher in bacteroids from determinate versus indeterminate nodules	[175]
*Rhizobium tropici* CIAT899^T^	Growth of wild type and *hss* mutant under salt stress	Slower growth of the mutant partially restored by 0.1 mM exogenous HSpd	HSpd important for salt stress resistance	[123]
*Sinorhizobium meliloti* (currently *Ensifer meliloti* [206]) strains Rm8530 and 1021	Phenotypic characterization of wild types and *nspS* null mutants	Altered biofilm formation, EPS production, and motility in presence of selected PAs, dependent on functionality of quorum sensing system	Exogenous PAs affect symbiotically relevant phenotypes in *S. meliloti*	[136]
*Sinorhizobium meliloti* (currently *Ensifer meliloti* [206]) Rm8530	Swimming motility of *hss cansdh* double mutant	Swimming significantly reduced versus wild type and *hss* and *cansdh* single mutants	Spd, HSpd, and/or NSpd are important for swimming motility	V. Becerra and M. Dunn, unpublished.
*Sinorhizobium meliloti* (currently *Ensifer meliloti* [206]) 1021	Wild type and *tolC* (outer membrane protein) mutant grown in cultures	*tolC* mutation provokes oxidative stress and higher expression of *odc1* and the Arg/Lys/Orn decarboxylase *sma0682*.	Cells of mutant attempt to counter oxidative stress by synthesizing more PAs	[114]
*Sinorhizobium meliloti* (currently *Ensifer meliloti* [206]) 1021	Inoculated on *M. truncatula* without or with salt stress and 0.l mM exogenous Spd or Spm	Treatments with Spd or Spm increased the expression of brassinosteroid synthesis genes and lowered oxidative stress provoked by high osmolarity	Cross-talk of PAs and the anti-stress phytoharmone brassinosteroid increase resistance to abiotic stress	[196]
*Sinorhizobium meliloti* (currently *Ensifer meliloti* [206]) GR4	In combination with *M. sativa* under salt stress without or with ABA pre-treatment	ABA pre-treatment increased Spd and Spm levels in nodules, increased antioxidant enzyme activities, and improved nitrogen fixation	ABA induces nodule antioxidant defenses mediated by Spd and Spm and other non-enzymatic antioxidants	[198]
*Sinorhizobium meliloti* (currently *Ensifer meliloti* [206]) Rm8530	Cultures with exogenous 1 mM Put, Spd or NSpd	Growth rate reduced up to 20% vs. control cultures	PAs moderately toxic at this concentration	[60]
*Sinorhizobium meliloti* (currently *Ensifer meliloti* [206]), various strains	Growth in culture or in symbiosis with *M. sativa* or *M. truncatula*	Sharply decreased intracellular PA content in *odc2* mutant or when Odc activity is inhibited with DFMO	PA biosynthesis required for wild type growth, (especially under abiotic stress) and full symbiotic efficiency. Chemical complementation with PAs or PA biosynthesis genes largely restores wild-type phenotypes	[60,62,82,83,115,207,208]
*Stenotrophomonas rhizophila* DSM14405^T^	Co-inoculated with *B. japonicum* on soybean	Synergistic effect that increases the symbiotic effectiveness of the nitrogen-fixing interaction under salt stress	Export of Spd or glucosylglycerol (an osmoprotectant) by DSM14405 may participate in plant growth promotion	[195]
*Stenotrophomonas rhizophila* DSM14405^T^	Growth with oilseed rape root exudates	Several-fold higher transcription of *mdtIJ* genes encoding a probable Spd export system	Spd exported by MdtIJ could promote plant growth and stress resistance	[193]
*Synechocystis* sp. PCC 6803	Growth under salt (NaCl) or osmotic (sorbitol) stress	Increased levels of PAs with osmotic shock, Adc mRNA stability greater under both stresses	PAs may balance ionic charges or increase Adc mRNA stability	[128]

## Figures and Tables

**Figure 1 plants-12-02671-f001:**
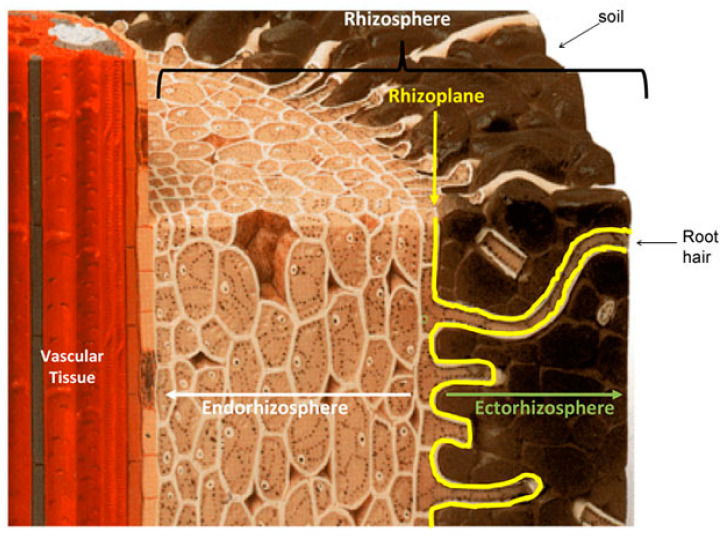
The rhizosphere can be divided into different parts. The innermost zone is known as the endorhizosphere, which encompasses both the interior of root cells and the apoplastic (extracellular) fluid-filled spaces surrounding them. Adjacent to the endorhizosphere is the rhizoplane, consisting of the root epidermis and the mucilage layer. Moving further outward, we encounter the ectorhizosphere, extending from the rhizoplane into the surrounding bulk soil. The accompanying figure was sourced from the Nature Education Knowledge Project (https://www.nature.com/scitable/knowledge/library/the-rhizosphere-roots-soil-and-67500617/, accessed on 9 March 2023).

**Figure 2 plants-12-02671-f002:**
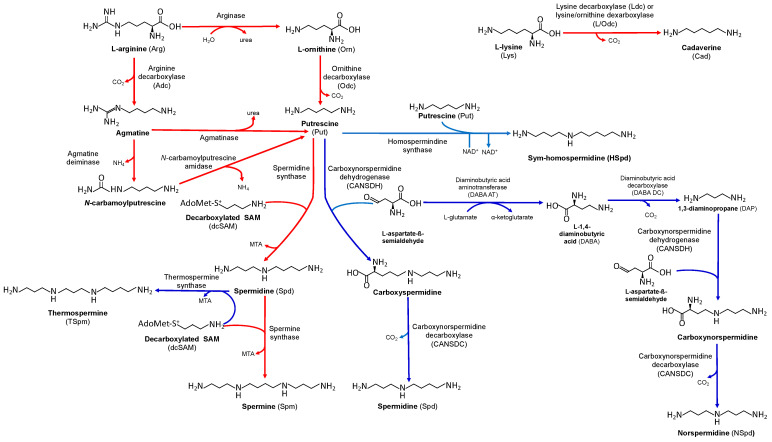
Polyamine biosynthesis pathways. Red arrows denote the canonical pathways present in plants and some bacteria, blue arrows indicate additional or alternative pathways present in some bacteria. Abbreviations not defined in the figure: MTA, 5′-methylthioadenosine; SAM, S-adenosyl methionine. The figure was made with Microsoft Powerpoint for Mac, version 16.74.

**Figure 3 plants-12-02671-f003:**
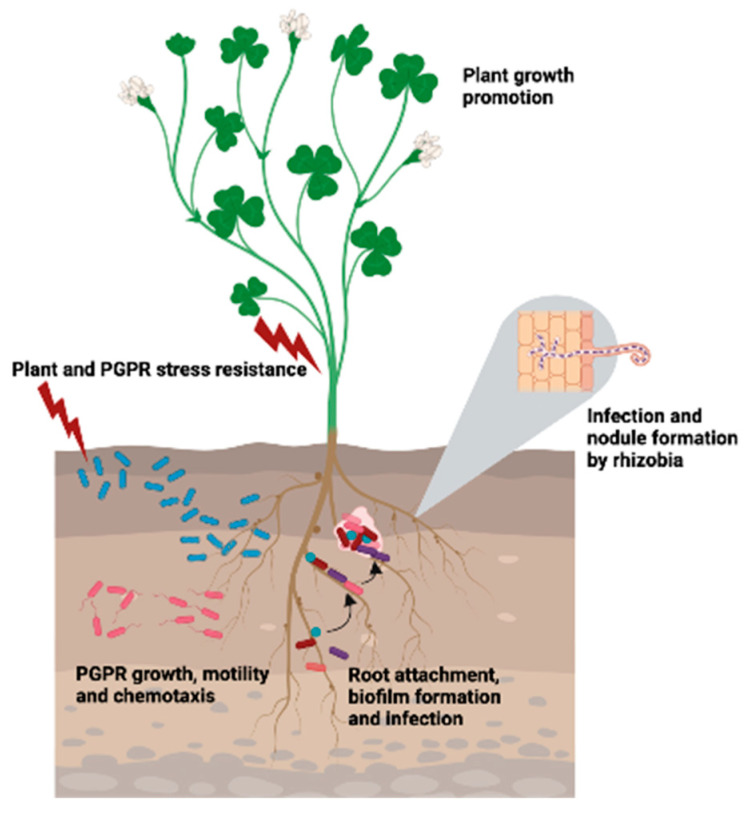
Processes influenced by polyamines in plant–PGPR interactions, with reference to relevant sections of this review. Clockwise from left: PAs in both plants and PGPR increase resistance to biotic and abiotic stress (Section 3.2.2, Section 4 and Section 5.2.2). Both endogenous PAs and those present in the environment influence PGPR growth, motility, and possibly chemotaxis (Section 3.2.1, Section 3.2.3 and Section 5.2.1). PAs are important or essential for root colonization and/or biofilm formation (Section 3.2.3). Nodule formation and symbiotic efficiency are significantly affected by PAs in both legumes and rhizobia (Section 5.2). The combined effects of PAs on these processes are an important factor in plant growth promotion. Created with BioRender.com.

## Data Availability

No new data were created or analyzed in this review.

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
