# Peer review of "The Biosynthesis and Functions of Polyamines in the Interaction of Plant Growth-Promoting Rhizobacteria with Plants"

_plants, 2023, doi:10.3390/plants12142671_

Round 1

Reviewer 1 Report

Dear Colleagues.

In my opinion, the manuscript «The biosynthesis and functions of polyamines in the interaction of plant growth-promoting rhizobacteria with plants» (Authors: Michael F. Dunn *, Victor Antonio Becerra-Rivera) is a very high-quality review on a current topic. The authors summarized a large number of recent studies on the role of polyamines in the regulation of processes in plant-microbial interactions. The data are summarized both for legume-rhizobium symbioses and the interaction of a wide range of plants with PGPR. The influence of polyamines on bacteria, plants, and plant-microbial associations is shown. The authors rightly note the difficulty of interpreting the role of polyamines themselves and their signaling functions. Data from literature sources were summarized mainly for the last 10 years. The review will be very useful to a wide range of researchers of plant-microbial symbioses.

I propose to improve the review with additional illustrative material. The review is illustrated with two figures representing the structure of the rhizosphere and the pathways of polyamine biosynthesis. It would be useful for readers if the authors of the review could present the signaling pathways in the formation of polyamines during the interaction of plants and PGPR, as well as their influence on macro- and microsymbionts.

Best regards,

Dr. O.V. Tkachenko

Author Response

Reviewer 1 comments

In my opinion, the manuscript «The biosynthesis and functions of polyamines in the interaction of plant growth-promoting rhizobacteria with plants» (Authors: Michael F. Dunn *, Victor Antonio Becerra-Rivera) is a very high-quality review on a current topic. The authors summarized a large number of recent studies on the role of polyamines in the regulation of processes in plant-microbial interactions. The data are summarized both for legume-rhizobium symbioses and the interaction of a wide range of plants with PGPR. The influence of polyamines on bacteria, plants, and plant-microbial associations is shown. The authors rightly note the difficulty of interpreting the role of polyamines themselves and their signaling functions. Data from literature sources were summarized mainly for the last 10 years. The review will be very useful to a wide range of researchers of plant-microbial symbioses.

I propose to improve the review with additional illustrative material. The review is illustrated with two figures representing the structure of the rhizosphere and the pathways of polyamine biosynthesis. It would be useful for readers if the authors of the review could present the signaling pathways in the formation of polyamines during the interaction of plants and PGPR, as well as their influence on macro- and microsymbionts.

We thank Dr. Tkachenko for his positive comments on the manuscript and his suggestion. We have included a summary of polyamine effects on PGPR and plants as Figure 3 in the revised version of the manuscript. However, the figure does not deal with signaling pathways because there is little data to suggest that PGPR directly or indirectly influence these pathways in plants. Thus, the figure is of a more general nature and concentrates on the effects of PAs on the symbiotic partners.

Reviewer 2 Report

The MS titled “The Biosynthesis and Functions of Polyamines in the Interaction of Plant Growth-Promoting Rhizobacteria with Plants” is detailed and comprehensive review article. MS has scientific relevance and quality to be published in "Plants". The title reflects the content of the article. The authors analyzed a large number of scientific sources on polyamines and plant-microbial interactions, with a significant part of them published in the last five years. This review is of scientific interest and practical significance, so MS will be useful for many scientists and students. Generally, the work is easy to understand. However, there are small remarks, which need to be addressed before its publication.

1.      It is advisable to provide a list of abbreviations, because there are a lot of them in MS.

2.      The caption under the Figure 1: The rhizosphere can be divided into different regions. Maybe it is better to use "parts" instead of "regions"?

3.      The Figure 2 that reflects polyamine biosynthesis pathways is a good illustration to review. However, since there is no reference, it can be concluded that it was compiled by the authors of MS. In this case, it is advisable to clarify in which program it was performed.

4.      There are some typos and inaccuracies in the text of MS:

P. 1; L.7: Mexio(?) instead of Mexico.

P. 4; L. 177: Indole-3-acetic acid is more correct than indole acetic acid.

P. 5; L. 190-191: In the amino acid names (L-arginine and L-ornithine) letter L is usually italicized.

There is no uniformity in the spelling of bacteria names: P. 8; L. 358: Anabaena sp. but in Table Anabaena sp. “sp.” in  italics.

P. 13; L. 385: reference 107 in curly bracket instead of square bracket.

            5. There are some inaccuracies in the “References”. For example:

in reference № 19 instead of the article number, the year of publication is indicated again;

the pages in the source № 51 are not very correctly indicated: 1276–88 instead of 1276–1288;

there is no indicate the volume number and pages in reference № 57.

Conclusion: I recommend the publication of the MS “The Biosynthesis and Functions of Polyamines in the Interaction of Plant Growth-Promoting Rhizobacteria with Plants”  in “Plants”  Journal after minor revision.

Author Response

Reviewer 2 comments

The MS titled “The Biosynthesis and Functions of Polyamines in the Interaction of Plant Growth-Promoting Rhizobacteria with Plants” is detailed and comprehensive review article. MS has scientific relevance and quality to be published in "Plants". The title reflects the content of the article. The authors analyzed a large number of scientific sources on polyamines and plant-microbial interactions, with a significant part of them published in the last five years. This review is of scientific interest and practical significance, so MS will be useful for many scientists and students. Generally, the work is easy to understand. However, there are small remarks, which need to be addressed before its publication.

  1. It is advisable to provide a list of abbreviations, because there are a lot of them in MS.

We thank the reviewer for his comments, error corrections and suggestions. We considered including an abbreviations list before submitting the paper, as we agree that there are a lot of abbreviations used in the manuscript. However, the Plants instructions to authors does not provide guidance on abbreviations, and none of the recent research papers or reviews published in Plants that we checked contain a list of abbreviations. For this reason, we have not included one but will do so at the request of the editor. Since most of the abbreviations in the manuscript deal with polyamine or enzyme names, the reader can consult Fig. 2, where both full names and abbreviations of these are given.

  1. The caption under the Figure 1: The rhizosphere can be divided into different regions. Maybe it is better to use "parts" instead of "regions"? This change was made.
  2. The Figure 2 that reflects polyamine biosynthesis pathways is a good illustration to review. However, since there is no reference, it can be concluded that it was compiled by the authors of MS. In this case, it is advisable to clarify in which program it was performed. This figure was done entirely with Microsoft Powerpoint for Mac, and this has been added to the figure legend.
  3. There are some typos and inaccuracies in the text of MS:
  4. 1; L.7: Mexio(?) instead of Mexico. Thanks, this was corrected.
  5. 4; L. 177: Indole-3-acetic acid is more correct than indole acetic acid. Agreed, the changed has been made.
  6. 5; L. 190-191: In the amino acid names (L-arginine and L-ornithine) letter L is usually italicized. We have change the L’s to small capitols, which we believe is the correct form for L-amino acid nomenclature according to the editorial board of J. Biol. Chem.

There is no uniformity in the spelling of bacteria names: P. 8; L. 358: Anabaena sp. but in Table Anabaena sp. “sp.” in  italics. The error in Table 1 was corrected.

  1. 13; L. 385: reference 107 in curly bracket instead of square bracket. This was corrected.
  2. There are some inaccuracies in the “References”. For example:

in reference № 19 instead of the article number, the year of publication is indicated again; This is how the journal handles the citations information for their papers, but we have included the article number from the doi in our citation.

the pages in the source № 51 are not very correctly indicated: 1276–88 instead of 1276–1288; Corrected.

there is no indicate the volume number and pages in reference № 57. This has been corrected.

Conclusion: I recommend the publication of the MS “The Biosynthesis and Functions of Polyamines in the Interaction of Plant Growth-Promoting Rhizobacteria with Plants”  in “Plants”  Journal after minor revision.

Reviewer 3 Report

This review describes the knowledge on the synthesis and the roles of polyamines (PAs) in plant growth-promoting rhizobacteria, in plants and in PGPR-plant interactions, especially under stress conditions. The topic is interesting, the authors have comprehensively reviewed the literature and the manuscript is well written. I only have some small suggestions to improve the manuscript.

The Introduction section could be shortened and be more focused. Moreover, repetition of some concepts could be avoided such as the description of the mechanisms underlying plant growth promotion by PGPR that are mentioned on lines 51-56 and on lines 143-145.

The position of Table 1 in the manuscript and of the references made to it are not the most logical. Table 1 is included in the middle of the manuscript (pp. 9-12) while it contains examples referred to in both sections 3.2 and 5.2. Instead of making references to Table 1 for a few examples such as the 3 mentions on page 8, I find it more logical to present Table 1 at the end of the manuscript and refer to it at the beginning of section 3.2 in a more general way as the authors did in the first paragraph of section 5.2.1.

 Azospirillum” should be corrected in Table 1 and reference 180

Author Response

Reviewer 3 comments

This review describes the knowledge on the synthesis and the roles of polyamines (PAs) in plant growth-promoting rhizobacteria, in plants and in PGPR-plant interactions, especially under stress conditions. The topic is interesting, the authors have comprehensively reviewed the literature and the manuscript is well written. I only have some small suggestions to improve the manuscript.

The Introduction section could be shortened and be more focused. Moreover, repetition of some concepts could be avoided such as the description of the mechanisms underlying plant growth promotion by PGPR that are mentioned on lines 51-56 and on lines 143-145.

We are thankful for the reviewer’s comments and suggestions. We believe that the Introduction section provides useful background to the topic of the review as well as the justification for writing it. At 111, lines, we don’t believe that its length is excessive and so have not modified it. With respect to the reviewer’s second point, the first mention (lines 51-56) of PGPR mechanisms is specific because we want to point out that the molecular signals involved in these phenomena are largely unknown. The second mention in lines 143-145 is included to indicate additional PGPR mechanisms in diazotrophs, besides nitrogen fixation. We believe that both of these sections serve a specific purpose and have not changed them.

The position of Table 1 in the manuscript and of the references made to it are not the most logical. Table 1 is included in the middle of the manuscript (pp. 9-12) while it contains examples referred to in both sections 3.2 and 5.2. Instead of making references to Table 1 for a few examples such as the 3 mentions on page 8, I find it more logical to present Table 1 at the end of the manuscript and refer to it at the beginning of section 3.2 in a more general way as the authors did in the first paragraph of section 5.2.1. We like this idea, and have made this change, including a reference to the Table at the beginning of Section 3.2.

 “Azospirillum” should be corrected in Table 1 and reference 180. These corrections were made.